# Simple MAP Inference via Low-Rank Relaxations

**Roy Frostig**[*],   **Sida I. Wang**,[*]   **Percy Liang,**   **Christopher D. Manning**
Computer Science Department, Stanford University, Stanford, CA, 94305
`{rf,sidaw,pliang}@cs.stanford.edu`, `manning@stanford.edu`

## Abstract

We focus on the problem of maximum *a posteriori* (MAP) inference in Markov random fields with binary variables and pairwise interactions. For this common subclass of inference tasks, we consider low-rank relaxations that interpolate between the discrete problem and its full-rank semidefinite relaxation. We develop new theoretical bounds studying the effect of rank, showing that as the rank grows, the relaxed objective increases but saturates, and that the fraction in objective value retained by the rounded discrete solution decreases. In practice, we show two algorithms for optimizing the low-rank objectives which are simple to implement, enjoy ties to the underlying theory, and outperform existing approaches on benchmark MAP inference tasks.

## 1   Introduction

Maximum *a posteriori* (MAP) inference in Markov random fields (MRFs) is an important problem with abundant applications in computer vision [1], computational biology [2], natural language processing [3], and others. To find MAP solutions, stochastic hill-climbing and mean-field inference are widely used in practice due to their speed and simplicity, but they do not admit any formal guarantees of optimality. Message passing algorithms based on relaxations of the marginal polytope [4] can offer guarantees (with respect to the relaxed objective), but require more complex bookkeeping. In this paper, we study algorithms based on low-rank SDP relaxations which are both remarkably simple and capable of guaranteeing solution quality.

Our focus is on MAP in a restricted but common class of models, namely those over binary variables coupled by pairwise interactions. Here, MAP can be cast as optimizing a quadratic function over the vertices of the $n$-dimensional hypercube: $\max_{x \in \{-1,1\}^n} x^\top A x$. A standard optimization strategy is to relax this integer quadratic program (IQP) to a semidefinite program (SDP), and then round the relaxed solution to a discrete one achieving a constant factor approximation to the IQP optimum [5, 6, 7]. In practice, the SDP can be solved efficiently using low-rank relaxations [8] of the form $\max_{X \in \mathbb{R}^{n \times k}} \operatorname{tr}(X^\top A X)$.

The first part of this paper is a theoretical study of the effect of the rank $k$ on low-rank relaxations of the IQP. Previous work focused on either using SDPs to solve IQPs [5] or using low-rank relaxations to solve SDPs [8]. We instead consider the direct link between the low-rank problem and the IQP. We show that as $k$ increases, the gap between the relaxed low-rank objective and the SDP shrinks, but vanishes as soon as $k \geq \operatorname{rank}(A)$; our bound adapts to the problem $A$ and can thereby be considerably better than the typical data-independent bound of $O(\sqrt{n})$ [9, 10]. We also show that the rounded objective shrinks in ratio relative to the low-rank objective, but at a steady rate of $\Theta(1/k)$ on average. This result relies on the connection we establish between IQP and low-rank relaxations. In the end, our analysis motivates the use of relatively small values of $k$, which is advantageous from both a solution quality and algorithmic efficiency standpoint.

---

[*]Authors contributed equally.

The second part of this paper explores the use of very low-rank *relaxation and randomized rounding* ($R^3$) in practice. We use projected gradient and coordinate-wise ascent for solving the $R^3$ relaxed problem (Section 4). We note that $R^3$ interfaces with the underlying problem in an extremely simple way, much like Gibbs sampling and mean-field: only a black box implementation of $x \mapsto Ax$ is required. This decoupling permits users to customize their implementation based on the structure of the weight matrix $A$: using GPUs for dense $A$, lists for sparse $A$, or much faster specialized algorithms for $A$ that are Gaussian filters [11]. In contrast, belief propagation and marginal polytope relaxations [2] need to track messages for each edge or higher-order clique, thereby requiring more memory and a finer-grained interface to the MRF that inhibits flexibility and performance.

Finally, we introduce a comparison framework for algorithms via the $x \mapsto Ax$ interface, and use it to compare $R^3$ with annealed Gibbs sampling and mean-field on a range of different MAP inference tasks (Section 5). We found that $R^3$ often achieves the best-scoring results, and we provide some intuition for our advantage in Section 4.1.

## 2  Setup and background

**Notation**   We write $\mathbb{S}_n$ for the set of symmetric $n \times n$ real matrices and $\mathcal{S}^k$ for the unit sphere $\{x \in \mathbb{R}^k : \|x\|_2 = 1\}$. All vectors are columns unless stated otherwise. If $X$ is a matrix, then $X_i \in \mathbb{R}^{1 \times k}$ is its $i$'th row.

This section reviews how MAP inference on binary graphical models with pairwise interactions can be cast as integer quadratic programs (IQPs) and approximately solved via semidefinite relaxations and randomized rounding. Let us begin with the definition of an IQP:

**Definition 2.1.** *Let $A \in \mathbb{S}_n$ be a symmetric $n \times n$ matrix. An (indefinite)* integer quadratic program *(IQP) is the following optimization problem:*

$$\max_{x \in \{-1,1\}^n} \quad \text{IQP}(x) \overset{\text{def}}{=} x^\mathsf{T} A x \tag{1}$$

Solving (1) is NP-complete in general: the MAX-CUT problem immediately reduces to it [5]. With an eye towards tractability, consider a first candidate relaxation: $\max_{x \in [-1,1]^n} x^\mathsf{T} A x$. This relaxation is *always* tight in that the maxima of the relaxed objective and original objective (1) are equal.[1] Therefore it is just as hard to solve. Let us then replace each scalar $x_i \in [-1, 1]$ with a unit vector $X_i \in \mathbb{R}^k$ and define the following *low-rank problem* (LRP):

**Definition 2.2.** *Let $k \in \{1, \dots, n\}$ and $A \in \mathbb{S}_n$. Define the* low-rank problem $\text{LRP}_k$ *by:*

$$\begin{aligned} \max_{X \in \mathbb{R}^{n \times k}} \quad & \text{LRP}_k(X) \overset{\text{def}}{=} \text{tr}(X^\mathsf{T} A X) \\ \text{subject to} \quad & \|X_i\|_2 = 1, \ i = 1, \dots, n. \end{aligned} \tag{2}$$

Note that setting $X_i = [x_i, 0, \dots, 0] \in \mathbb{R}^k$ recovers (1). More generally, we have a sequence of successively looser relaxations as $k$ increases. What we get in return is tractability. The $\text{LRP}_k$ objective generally yields a non-convex problem, but if we take $k = n$, the objective can be rewritten as $\text{tr}(X^\mathsf{T} A X) = \text{tr}(A X X^\mathsf{T}) = \text{tr}(A S)$, where $S$ is a positive semidefinite matrix with ones on the diagonal. The result is the classic SDP relaxation, which is convex:

$$\begin{aligned} \max_{S \in \mathbb{S}_n} \quad & \text{SDP}(S) \overset{\text{def}}{=} \text{tr}(A S) \\ \text{subject to} \quad & S \succeq 0, \ \text{diag}(S) = \mathbf{1} \end{aligned} \tag{3}$$

Although convexity begets easy optimization in a theoretical sense, the number of variables in the SDP is quadratic in $n$. Thus for large SDPs, we actually return to the low-rank parameterization (2). Solving $\text{LRP}_k$ via simple gradient methods works extremely well in practice and is partially justified by theoretical analyses in [8, 12].

To complete the picture, we need to convert the relaxed solutions $X \in \mathbb{R}^{n \times k}$ into integral solutions $x \in \{-1, 1\}^n$ of the original IQP (1). This can be done as follows: draw a vector $g \in \mathbb{R}^k$ on the unit sphere uniformly at random, project each $X_i$ onto $g$, and take the sign. Formally, we write $x = \mathrm{rrd}(X)$ to mean $x_i = \mathrm{sign}(X_i \cdot g)$ for $i = 1, \ldots, n$. This randomized rounding procedure was pioneered by [5] to give the celebrated $0.878$-approximation of MAX-CUT.

## 3 Understanding the relaxation-rounding tradeoff

The overall IQP strategy is to first relax the integer problem domain, then round back in to it. The optimal objective increases in relaxation, but decreases in randomized rounding. How do these effects compound? To guide our choice of relaxation, we analyze the effect that the rank $k$ in (2) has on the *approximation ratio* of rounded versus optimal IQP solutions.

More formally, let $x^\star$, $X^\star$, and $S^\star$ denote global optima of IQP, of $\mathrm{LRP}_k$, and of SDP, respectively. We can decompose the approximation ratio as follows:

$$1 \geq \underbrace{\frac{\mathbb{E}[\mathrm{IQP}(\mathrm{rrd}(X^\star))]}{\mathrm{IQP}(x^\star)}}_{\text{approximation ratio}} = \underbrace{\frac{\mathrm{SDP}(S^\star)}{\mathrm{IQP}(x^\star)}}_{\text{constant} \geq 1} \times \underbrace{\frac{\mathrm{LRP}_k(X^\star)}{\mathrm{SDP}(S^\star)}}_{\text{tightening ratio } T(k)} \times \underbrace{\frac{\mathbb{E}[\mathrm{IQP}(\mathrm{rrd}(X^\star))]}{\mathrm{LRP}_k(X^\star)}}_{\text{rounding ratio } R(k)} \tag{4}$$

As $k$ increases from 1, the tightening ratio $T(k)$ increases towards 1 and the rounding ratio $R(k)$ decreases from 1. In this section, we lower bound $T$ and $R$ each in turn, thus lower-bounding the approximation ratio as a function of $k$. Specifically, we show that $T(k)$ reaches 1 at small $k$ and that $R(k)$ falls as $\frac{2}{\pi} + \Theta(\frac{1}{k})$.

In practice, one cannot find $X^\star$ for general $k$ with guaranteed efficiency (if we could, we would simply use $\mathrm{LRP}_1$ to directly solve the original IQP). However, Section 5 shows empirically that simple procedures solve $\mathrm{LRP}_k$ well for even small $k$.

### 3.1 The tightening ratio $T(k)$ increases

We now show that, under the assumption of $A \succeq 0$, the tightening ratio $T(k)$ plateaus early and that it approaches this plateau steadily. Hence, provided $k$ is beyond this saturation point, and large enough so that an $\mathrm{LRP}_k$ solver is practically capable of providing near-optimal solutions, there is no advantage in taking $k$ larger.

First, $T(k)$ is *steadily* bounded below. The following is a result of [13] (that also gives insight into the theoretical worst-case hardness of optimizing $\mathrm{LRP}_k$):

**Theorem 3.1** ([13]). *Fix $A \succeq 0$ and let $S^\star$ be an optimal* SDP *solution. There is a randomized algorithm that, given $S^\star$, outputs $\tilde{X}$ feasible for $\mathrm{LRP}_k$ such that $\mathbb{E}_{\tilde{X}}[\mathrm{LRP}_k(\tilde{X})] \geq \gamma(k) \cdot \mathrm{SDP}(S^\star)$, where*

$$\gamma(k) \overset{\text{def}}{=} \frac{2}{k} \left( \frac{\Gamma((k+1)/2)}{\Gamma(k/2)} \right)^2 = 1 - \frac{1}{2k} + o\left( \frac{1}{k} \right) \tag{5}$$

*For example, $\gamma(1) = \frac{2}{\pi} = 0.6366$, $\gamma(2) = 0.7854$, $\gamma(3) = 0.8488$, $\gamma(4) = 0.8836$, $\gamma(5) = 0.9054$.*[2]

By optimality of $X^\star$, $\mathrm{LRP}_k(X^\star) \geq \mathbb{E}_{\tilde{X}}[\mathrm{LRP}_k(\tilde{X})]$ under any probability distribution, so the existence of the algorithm in Theorem 3.1 implies that $T(k) \geq \gamma(k)$.

Moreover, $T(k)$ achieves its maximum of 1 at small $k$, and hence must strictly exceed the $\gamma(k)$ lower bound early on. We can arrive at this fact by bounding the rank of the SDP-optimal solution $S^\star$. This is because $S^\star$ factors into $S^\star = XX^\mathsf{T}$, where $X$ is in $\mathbb{R}^{n \times \mathrm{rank}\, S^\star}$ and must be optimal since $\mathrm{LRP}_{\mathrm{rank}\, S^\star}(X) = \mathrm{SDP}(S^\star)$. Without consideration of $A$, the following theorem uniformly bounds this rank at well below $n$. The theorem was established independently by [9] and [10]:

**Theorem 3.2** ([9, 10]). *Fix a weight matrix $A$. There exists an optimal solution $S^\star$ to* SDP *(3) such that* $\mathrm{rank}\, S^\star \leq \sqrt{2n}$.

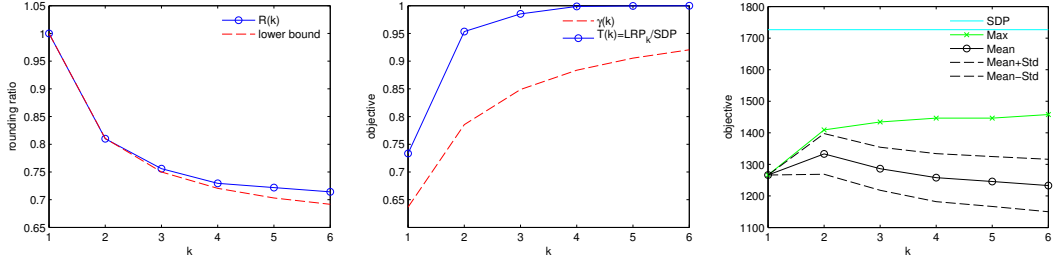

(a) $R(k)$ (blue) is close to it $2/(\pi\gamma(k))$ lower bound (red) across the small $k$.

(b) $\tilde{T}(k)$ (blue), the empirical tightening ratio, clears its lower bound $\gamma(k)$ (red) and hits its ceiling of 1 at $k = 4$.

(c) Rounded objective values vs. $k$: optimal SDP (cyan), best IQP rounding (green), and mean IQP rounding $\pm\sigma$ (black).

Figure 1: Plots of quantities analyzed in Section 3, under $A \in \mathbb{R}^{100 \times 100}$ whose entries are sampled independently from a unit Gaussian. For this instance, the empirical post-rounding objectives are shown at the right for completeness.

Hence we know already that the tightening ratio $T(k)$ equals 1 by the time $k$ reaches $\sqrt{2n}$.

Taking $A$ into consideration, we can identify a class of problem instances for which $T(k)$ actually saturates at even smaller $k$. This result is especially useful when the rank of the weight matrix $A$ is known, or even under one's control, while modeling the underlying optimization task:

**Theorem 3.3.** *If $A$ is symmetric, there is an optimal* SDP *solution $S^\star$ such that* $\operatorname{rank} S^\star \leq \operatorname{rank} A$.

A complete proof is in Appendix A.1. Because adding to the diagonal of $A$ is equivalent to merely adding a constant to the objective of all problems considered, Theorem 3.3 can be strengthened:

**Corollary 3.4.** *For any symmetric weight matrix $A$, there exists an optimal* SDP *solution $S^\star$ such that* $\operatorname{rank} S^\star \leq \min_{u \in \mathbb{R}^n} \operatorname{rank}(A + \operatorname{diag}(u))$.

That is, changes to the diagonal of $A$ that reduce its rank may be applied to improve the bound.

In summary, $T(k)$ grows at least as fast as $\gamma(k)$, from $T(k) = 0.6366$ at $k = 1$ to $T(k) = 1$ at $k = \min\{\sqrt{2n}, \min_{u \in \mathbb{R}^n} \operatorname{rank}(A + \operatorname{diag}(u))\}$. This is validated empirically in Figure 1b.

### 3.2 The rounding ratio $R(k)$ decreases

As the dimension $k$ grows for row vectors $X_i$ in the LRP$_k$ problem, the rounding procedure incurs a larger expected drop in objective value. Fortunately, we can bound this drop. Even more fortunately, the bound grows no faster than $\gamma(k)$, exactly the *steady* lower bound for $T(k)$. We obtain this result with an argument based on the analysis of [13]:

**Theorem 3.5.** *Fix a weight matrix $A \succeq 0$ and any* LRP$_k$*-feasible $X \in \mathbb{R}^{n \times k}$. The rounding ratio for $X$ is bounded below as*

$$\frac{\mathbb{E}[\mathrm{IQP}(\mathrm{rrd}(X))]}{\mathrm{LRP}_k(X)} \geq \frac{2}{\pi\gamma(k)} = \frac{2}{\pi}\left(1 + \frac{1}{2k} + o\left(\frac{1}{k}\right)\right) \tag{6}$$

Note that $X$ in the theorem need not be optimal – the bound applies to whatever solution an LRP$_k$ solver might provide. The proof, given in Appendix section A.1, uses Lemma 1 from [13], which is based on the theory of positive definite functions on spheres [14]. A decrease in $R(k)$ that tracks the lower bound is observed empirically in Figure 1a.

In summary, considering only the steady bounds (Theorems 3.1 and 3.5), $T$ will always rise opposite to $R$ at least at the same rate. Then, the added fact that $T$ plateaus early (Theorem 3.2 and Corollary 3.4) means that $T$ in fact rises even faster.

In practice, we would like to take $k$ beyond 1 as we find that the first few relaxations give the optimizer an increasing advantage in arriving at a good LRP$_k$ solution, close to $X^\star$ in objective. The rapid rise of $T$ relative to $R$ just shown then justifies not taking $k$ much larger if at all.

## 4   Pairwise MRFs, optimization, and inference alternatives

Having understood theoretically how IQP relates to low-rank relaxations, we now turn to MAP inference and empirical evaluation. We will show that the $\text{LRP}_k$ objective can be optimized via a simple interface to the underlying MRF. This interface then becomes the basis for (a) a MAP inference algorithm based on very low-rank relaxations, and (b) a comparison to two other basic algorithms for MAP: Gibbs sampling and mean-field variational inference.

A binary pairwise Markov random field (MRF) models a function $h$ over $x \in \{0,1\}^n$ given by $h(x) = \sum_i \psi_i(x_i) + \sum_{i<j} \theta_{i,j}(x_i, x_j)$, where the $\psi_i$ and $\theta_{i,j}$ are real-valued functions. The MAP inference problem asks for the variable assignment $x^\star$ that maximizes the function $h$. An MRF being binary-valued and pairwise allows the arbitrary factor tables $\psi_i$ and $\theta_{i,j}$ to be transformed with straightforward algebra into weights $A \in \mathbb{S}_n$ for the IQP. For the complete reduction, see Appendix A.2.

We make Section 3 actionable by defining the *randomized relaxation and rounding* ($\text{R}^3$) algorithm for MAP via low-rank relaxations. The first step of this algorithm involves optimizing $\text{LRP}_k$ (2) whose weight matrix encodes the MRF. In practice, MRFs usually have special structure, e.g., edge sparsity, factor templates, and Gaussian filters [11]. To develop $\text{R}^3$ as a general tool, we provide two interfaces between the solver and MRF representation, both of which allow users to exploit special structure.

**Left-multiplication ($x \mapsto Ax$)**   Assume a function $F$ that implements left matrix multiplication by the MRF matrix $A$. This suffices to compute the gradient of the relaxed objective: $\nabla_X \text{LRP}_k(X) = 2AX$. We can optimize the relaxation using projected gradient ascent (PGA): alternate between taking gradient steps and projecting back onto the feasible set (unit-normalizing the rows $X_i$ if the norm exceeds 1); see Algorithm 1. A user supplying a left-multiplication routine can parallelize its implementation on a GPU, use sparse linear algebra, or efficiently implement a dense filter.

**Row-product ($(i,x) \mapsto A_i x$)**   If the function $F$ further provides left multiplication by any row of $A$, we can optimize $\text{LRP}_k$ with coordinate-wise ascent (BCA). Fixing all but the $i$'th row of $X$ gives a function linear in $X_i$ whose optimum is $A_i X$ normalized to have unit norm.

Left-multiplication is suitable when one expects to parallelize multiplication, or exploit common dense structure as with filters. Row product is suitable when one already expects to compute $Ax$ serially. BCA also eliminates the need for the step size scheme in PGA, thus reducing the number of calls to the left-multiplication interface if this step size is chosen by line search.

---

$X \leftarrow$ random initialization in $\mathbb{R}^{k \times n}$
**for** $t \leftarrow 1$ **to** $T$ **do**
    **if** *parallel* **then**
        $X \leftarrow \Pi_{\mathcal{S}^k}(X + 2\eta_t AX)$    // Parallel update
    **else**
        **for** $i \leftarrow 1$ **to** $n$ **do**
            $X_i \leftarrow \Pi_{\mathcal{S}^k}(\langle A_i, X \rangle)$    // Sweep update
**for** $j \leftarrow 1$ **to** $M$ **do**
    $x^{(j)} \leftarrow \text{sign}(Xg)$, where $g$ is a random vector from unit sphere $\mathcal{S}^k$ (normalized Gaussian)
Output the $x^{(j)}$ for which the objective $(x^{(j)})^\mathsf{T} A x^{(j)}$ is largest.

**Algorithm 1:** The full randomized relax-and-round ($\text{R}^3$) procedure, given a weight matrix $A$; $\Pi_{\mathcal{S}^k}(\cdot)$ is row normalization and $\eta_t$ is the step size in the $t$'th iteration.

---

### 4.1   Comparison to Gibbs sampling and mean-field

The $\text{R}^3$ algorithm affords a tidy comparison to two other basic MAP algorithms. First, it is iterative and maintains a constant amount of state per MRF variable (a length $k$ row vector). Using the row-product interface, $\text{R}^3$ under BCA sequentially sweeps through and updates each variable's state (row $X_i$) while holding all others fixed. This interface bears a striking resemblance to (annealed) Gibbs sampling and mean-field iterative updates [4, 15], which are popular due to their simplicity. Table 1 shows how both can be implemented via the row-product interface.

| Algorithm | Domain | Sweep update | Parallel update |
|---|---|---|---|
| Gibbs | $x \in \{-1, 1\}^n$ | $x_i \sim \Pi_Z(\exp(A_i x))$ | $x \sim \Pi_Z(\exp(Ax))$ |
| Mean-field | $x \in [-1, 1]^n$ | $x_i \leftarrow \tanh(A_i x)$ | $x \leftarrow \tanh(Ax)$ |
| R$^3$ | $X \in (\mathcal{S}^k)^n$ | $X_i \leftarrow \Pi_{\mathcal{S}^k}(A_i X)$ | $X \leftarrow \Pi_{\mathcal{S}^k}(X + 2\eta_t AX)$ |

Table 1: Iterative updates for MAP algorithms that use constant state per MRF variable. $\Pi_{\mathcal{S}^k}$ denotes $\ell_2$ unit-normalization of rows and $\Pi_Z$ denotes scaling rows so that they sum to 1. The R$^3$ sweep update is not a gradient step, but rather the analytic maximum for the $i$'th row fixing the rest.

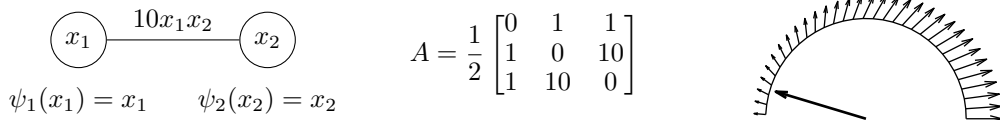

$\psi_1(x_1) = x_1 \qquad \psi_2(x_2) = x_2$

$A = \dfrac{1}{2} \begin{bmatrix} 0 & 1 & 1 \\ 1 & 0 & 10 \\ 1 & 10 & 0 \end{bmatrix}$

Figure 2: Consider the two variable MRF on the left (with $x_1, x_2 \in \{-1, 1\}$ for the factor expressions) and its corresponding matrix $A$. Note $x_0$ is clamped to 1 as per the reduction (A.2). The optimum is $x = [1, 1, 1]^\mathsf{T}$ with a value of $x^\mathsf{T} A x = 12$. If Gibbs or LRP$_1$ is initialized at $x = [1, -1, -1]^\mathsf{T}$, then either one will be unlikely to transition away from its suboptimal objective value of $8$ (as flipping only one of $x_1$ or $x_2$ decreases the objective to $-10$). Meanwhile, LRP$_2$ succeeds with probability 1 over random initializations. Suppose $X = [1, 0; X_1; X_2]$ with $X_1 = X_2$. Then the gradient update is $X_1 = \Pi_{\mathcal{S}^2}(A_1 X) = \Pi_{\mathcal{S}^2}(([1, 0] + 10X_1)/2)$, which always points towards $X_1^\star = X_2^\star = [1, 0]$ except in the 0-probability event that $X_1 = X_2 = [-1, 0]$ (corresponding to the poor initialization of $[1, -1, -1]^\mathsf{T}$ above). The gradient with respect to $X_1$ at points along the unit circle is shown on the right. The thick arrow represents an $X_1 \approx [-0.95, 0.3]$, and the gradient field shows that it will iteratively update towards the optimum.

Using left-multiplication, R$^3$ updates the state of all variables in parallel. Superficially, both Gibbs and the iterative mean-field update can be parallelized in this way as well (Table 1), but doing so incorrectly alters the their convergence properties. Nonetheless, [11] showed that a simple modification works well in practice for mean-field, so we consider these algorithms for a complete comparison.[3]

While Gibbs, mean-field, and R$^3$ are similar in form, they differ in their per-variable state: Gibbs maintains a number in $\{-1, 1\}$ whereas R$^3$ stores an entire vector in $\mathbb{R}^k$. We can see by example that the extra state can help R$^3$ avoid local optima that ensnarls Gibbs. A single coupling edge in a two-node MRF, described in Figure 2, gives intuition for the advantage of optimizing relaxations over stochastic hill-climbing.

Another widely-studied family of MAP inference techniques are based on belief propagation or relaxations of the marginal polytope [4]. For belief propagation, and even for the most basic of the LP relaxations (relaxing to the local consistency polytope), one needs to store state for every edge in addition to every variable. This demands a more complex interface to the MRF, introduces substantial added bookkeeping for dense graphs, and is not amenable to techniques such as the filter of [11].

## 5   Experiments

We compare the algorithms from Table 1 on three benchmark MRFs and an additional artificial MRF. We also show the effect of the relaxation $k$ on the benchmarks in Figure 3.

**Rounding in practice**    The theory of Section 3 provides safeguard guarantees by considering the average-case rounding. In practice, we do far better than average since we take several roundings and output the best. Similarly, Gibbs' output is taken as the best along its chain.

**Budgets**    Our goal is to see how efficiently each method utilizes the same fixed *budget* of queries to the function, so we fix the number queries to the left-multiplication function $F$ of Section 4. A budget jointly limits the relaxation updates and the number of random roundings taken in R$^3$. We charge

| | | algo. | dataset [name (# of instances)] | | | | | | | |
|---|---|---|---|---|---|---|---|---|---|---|
| | | | seg (50) | | dbn (108) | | grid40 (8) | | chain (300) | |
| low budget | sweep | Gibbs | 8.35 | (23) | 1.39 | (30) | **14.5** | **(7)** | .473 | (37) |
| | | MF | 8.36 | (23) | 1.3 | (7) | 13.6 | (1) | .463 | (39) |
| | | $R^3$ | 8.39 | (15) | **1.42** | **(71)** | 13.7 | (0) | **.538** | **(296)** |
| | parallel | Gibbs | 7.4 | (19) | .826 | (3) | .843 | (0) | .124 | (3) |
| | | MF | 7.4 | (26) | 1.16 | (6) | 11.3 | (3) | .35 | (50) |
| | | $R^3$ | 7.4 | (17) | **1.29** | **(99)** | 11.3 | (5) | **.418** | **(282)** |
| high budget | sweep | Gibbs | 7.07 | (33) | 1.26 | (42) | **12.5** | **(7)** | .367 | (85) |
| | | MF | 7.03 | (9) | 1.16 | (4) | 11.7 | (1) | .33 | (39) |
| | | $R^3$ | 7.09 | (23) | **1.28** | **(62)** | 11.9 | (0) | **.398** | **(300)** |
| | parallel | Gibbs | 6.78 | (31) | .814 | (2) | 1.85 | (0) | .132 | (11) |
| | | MF | 6.75 | (12) | 1.1 | (2) | 10.9 | (2) | .259 | (47) |
| | | $R^3$ | 6.8 | (25) | **1.25** | **(104)** | **11** | **(6)** | **.321** | **(296)** |

Table 2: Benchmark performance of algorithms in each comparison regime, in which the benchmarks are held to different computational budgets that cap their access to the left-multiplication routine. The score shown is an average relative gain in objective over the uniform-random baseline. Parenthesized is the win count (including ties), and bold text highlights qualitatively notable successes.

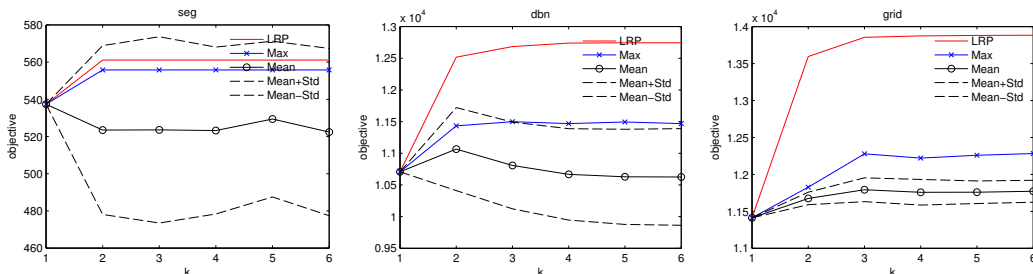

Figure 3: Relaxed and rounded objectives vs. the rank $k$ in an instance of **seg**, **dbn**, and **grid40**. Blue: max of roundings. Red: value of $LRP_k$. Black: mean of roundings ($\pm\sigma$). The relaxation objective increases with $k$, suggesting that increasingly good solutions are obtained by increasing $k$, in spite of non-convexity (here we are using parallel updates, *i.e.* using $R^3$ with PGA). The maximum rounding also improves considerably with $k$, especially at first when increasing beyond $k = 1$.

$R^3$ $k$-fold per use of $F$ when updating, as it queries $F$ with a $k$-row argument.[4] Sweep methods are charged once per pass through all variables.

We experiment with separate budgets for the sweep and parallel setup, as sweeps typically converge more quickly. The benchmark is run under separate low and high budget regimes – the latter more than double the former to allow for longer-run effects to set in. In Table 2, the sweep algorithms' low budget is 84 queries; the high budget is 200. The parallel low budget is 180; the high budget is 400. We set $R^3$ to take 20 roundings under low budgets and 80 under high ones, and the remaining budget goes towards $LRP_k$ updates.

**Datasets** Each dataset comprises a family of binary pairwise MRFs. The sets **seg**, **dbn**, and **grid40** are from the PASCAL 2011 Probabilistic Inference Challenge[5] — **seg** are small segmentation models (50 instances, average 230 variables, 622 edges), **dbn** are deep belief networks (108 instances, average 920 variables, 54160 edges), and **grid40** are 40x40 grids (8 instances, 1600 variables, 6240 or 6400 edges) whose edge weights outweigh their unaries by an order of magnitude. The **chain** set comprises 300 randomly generated 20-node chain MRFs with no unary potentials and random unit-Gaussian edge weights – it is principally an extension of the coupling two-node example (Figure 2), and serves as a structural obverse to **grid40** in that it lacks cycles entirely. Among these, the **dbn** set comprises the largest and most edge-dense instances.

**Evaluation**     To aggregate across instances of a dataset, we measure the average improvement over a simple baseline that, subject to the budget constraint, draws uniformly random vectors in $\{-1, 1\}^n$ and selects the highest-scoring among them. Improvement over the baseline is relative: if $z$ is the solution objective and $z'$ is that of the baseline, $(z - z')/z'$ is recorded for the average. We also count wins (including ties), the number of times a method obtains the best objective among the competition. Baseline performance varies with budget so scores are incomparable across sweep and parallel experiments.

In all experiments, we use LRP$_4$, *i.e.* the width-4 relaxation. The R$^3$ gradient step size scheme is $\eta_t = 1/\sqrt{t}$. In the parallel setting, mean-field updates are prone to large oscillations, so we smooth the update with the current point: $x \leftarrow (1 - \eta)x + \eta \tanh(Ax)$. Our experiments set $\eta = 0.5$. Gibbs is annealed from an initial temperature of 10 down to 0.1. These settings were tuned towards the benchmarks using a few arbitrary instances from each dataset.

Results are summarized in Table 2. All methods fare well on the **seg** dataset and find solutions very near the apparent global optimum. This shows that the rounding scheme of R$^3$, though elementary, is nonetheless capable of recovering an actual MAP point. On **grid40**, R$^3$ is competitive but not outstanding, and on **chain** it is a clear winner. Both datasets have edge potentials that dominate the unaries, but the cycles in the grid help break local frustrations that occur in **chain** where they prevents Gibbs from transitioning. On **dbn** – the more difficult task grounded in a real model – R$^3$ outperforms the others by a large margin.

Figure 3 demonstrates that relaxation beyond the quadratic program $\max_{x \in [-1,1]} x^\mathsf{T} Ax$ (*i.e.* $k = 1$) is crucial, both for optimizing LRP$_k$ and for obtaining a good maximum among roundings. Figure 4 in the appendix visualizes the distribution of rounded objective values across different instances and relaxations, illustrating that the difficulty of the problem can be apparent in the rounding distribution.

# 6   Related work and concluding remarks

In this paper, we studied MAP inference problems that can be cast as an integer quadratic program over hypercube vertices (IQP). Relaxing the IQP to an SDP (3) and rounding back with $\mathrm{rrd}(\cdot)$ was introduced by Goemans and Williamson in the 1990s for MAX-CUT. It was generalized to positive semidefinite weights shortly thereafter by Nesterov [6].

Separately, in the early 2000s, there was interest in scalably solving SDPs, though not with the specific goal of solving the IQP. The low-rank reparameterization of an SDP, as in (2), was developed by [8] and [12]. Recent work has taken this approach to large-scale SDP formulations of clustering, embedding, matrix completion, and matrix norm optimization for regularization [17, 18]. Upper bounds on SDP solutions in terms of problem size $n$, which help justify using a low rank relaxation, have been known since the 1990s [9, 10].

The natural joint use of these ideas (IQP relaxed to SDP and SDP solved by low-rank relaxation) is somewhat known. It was applied in a clustering experiment in [19], but no theoretical analysis was given and no attention paid to rounding directly from a low-rank solution. The benefit of rounding from low-rank was noticed in coarse MAP experiments in [20], but no theoretical backing was given and no attention paid to coordinate-wise ascent or budgeted queries to the underlying model.

Other relaxation hierarchies have been studied in the MRF MAP context, namely linear program (LP) relaxations given by hierarchies of outer bounds on the marginal polytope [21, 2]. They differ from this paper's setting in that they maintain state for every MRF clique configuration – an approach that extends beyond pairwise MRFs but that scales with the number of factors (unwieldy versus a large, dense binary pairwise MRF) and requires fine-grained access to the MRF. Sequences of LP and SDP relaxations form the Sherali-Adams and Lasserre hierarchies, respectively, whose relationship is discussed in [4] (Section 9). The LRP$_k$ hierarchy sits at a lower level: between the IQP (1) and the first step of the Lasserre hierarchy (the SDP (3)).

From a practical point of view, we have presented an algorithm very similar in form to Gibbs sampling and mean-field. This provides a down-to-earth perspective on relaxations within the realm of scalable and simple inference routines. It would be interesting to see if the low-rank relaxation ideas from this paper can be adapted to other settings (*e.g.*, for marginal inference). Conversely, the rich literature on the Lasserre hierarchy may offer guidance in extending the low-rank semidefinite approach (*e.g.*, beyond the binary pairwise setting).

## Footnotes

[1]*Proof.* WLOG, $A \succeq 0$ because adding to its diagonal merely adds a constant term to the IQP objective. The objective is a convex function, as we can factor $A = LL^\mathsf{T}$ and write $x^\mathsf{T} LL^\mathsf{T} x = \|L^\mathsf{T} x\|_2^2$, so it must be maximized over its convex polytope domain at a vertex point. $\qquad \square$

[2]The function $\gamma(k)$ generalizes the constant approximation factor $2/\pi = \gamma(1)$ with regards to the implications of the unique games conjecture: the authors show that no polynomial time algorithm can, in general, approximate $\mathrm{LRP}_k$ to a factor greater than $\gamma(k)$ assuming P $\neq$ NP and the UGC.

[3]Later, in [16], the authors derive the parallel mean-field update as being that of a concave approximation to the cross-entropy term in the true mean-field objective.

[4] This conservatively disfavors $R^3$, as it ignores the possible speedups of treating length-$k$ vectors as a unit.

[5] `http://www.cs.huji.ac.il/project/PASCAL/`

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
