[Supplementary Material]

# A   Appendix

## A.1   Proofs

### A.1.1   Proof of Theorem 3.3

*Proof.*   For simplicity, we first consider the case of symmetric PSD $A$. Let $k^\star = \operatorname{rank} A$. Consider $X \in \mathbb{R}^{n \times k}$ with $||X_i||_2 \leq 1$ and $k > k^\star$ such that $\operatorname{LRP}_k(X) = \operatorname{tr}(X^\mathsf{T} A X)$ attains the optimal value of the SDP (this is possible in particular when $k = n$). We want to to transform $X$ to the thinner $X^\star \in \mathbb{R}^{n \times k^\star}$ that still satisfies the row norm constraints $||X_i^\star||_2 \leq 1$. Let $Q \in \mathbb{R}^{k \times k}$ be an orthonormal matrix ($QQ^\mathsf{T} = I_k$). Note that $XQ$ still satisfies the row norm constraints (since each row of $X_i$ just gets rotated). Thus, it suffices to find $Q$ so that some columns of $XQ$ fall into the null-space of $A$ and can be discarded.

Suppose $A \succeq 0$. Let $A = LL^\mathsf{T}$ for $L \in \mathbb{R}^{n \times k^\star}$ and let $Y = L^\mathsf{T} X \in \mathbb{R}^{k^\star \times k}$. We can choose $Q$ so that $YQ \in \mathbb{R}^{k^\star \times k}$ has at most $k^\star$ non-zero columns, *i.e.* take $Q = [Q_\text{basis}, Q_\text{null}]$, where $Q_\text{null} \in \mathbb{R}^{k \times (k-k^\star)}$ comprises the $k - k^\star$ columns such that $YQ_\text{null} = 0$ and $Q_\text{basis} \in \mathbb{R}^{k \times k^\star}$ comprises the first $k^\star$ columns of $Q$. Obtaining such a $Q$ is possible by taking an orthonormal basis of the null space of $Y$ as the columns of $Q_\text{null}$, and taking an orthonormal basis of the $k^\star$-dimensional row space of $Y$ as the columns of $Q_\text{basis}$. Both bases can be obtained by applying the Gram-Schmidt process.

Now when we transform $X$ by $Q$ to get $XQ = [XQ_\text{basis}, XQ_\text{null}]$, we can drop the columns $XQ_\text{null}$ since $0 = YQ_\text{null} = L^\mathsf{T} XQ_\text{null}$, thus removing $XQ_\text{null}$ does not change the objective. Setting $X^\star = XQ_\text{basis} \in \mathbb{R}^{n \times k^\star}$ gives that $\operatorname{LRP}_k(X^\star) = \operatorname{LRP}_k(X)$ and we get the desired rank reduction without changing the objective and while maintaining satisfiability of the row norm constraints.

More generally if $A$ is real symmetric (but not necessarily $A \succeq 0$) then we can consider instead the factorization $A = LR^\mathsf{T}$ where the columns of $R$ are identical to the columns of $L$ except possibly negated. Such a factorization is given by the eigendecomposition of a real symmetric matrix. In this case, $Q$ still rotates both $L$ and $R$ correctly and the above argument follows in the same way. $\qquad\square$

We remark that even more generally, if $A = LU^\mathsf{T}$ for $L, U \in \mathbb{R}^{n \times k^\star}$ for $n \geq k \geq 2k^\star$, then we can set $Q_\text{basis}$ to be the basis of the row space of $Y = [L^\mathsf{T} X; U^\mathsf{T} X] \in \mathbb{R}^{2k^\star \times k}$. Then the same argument still applies but we can only reduce the solution rank from $k$ to $2k^\star = 2\operatorname{rank}(A)$.

### A.1.2   Proof of Theorem 3.5

*Proof.*   The proof relies on Grothendieck's identity: if $u, v \in \mathbb{R}^k$ and $g$ is drawn uniformly from the unit sphere $\mathcal{S}^k$, then

$$\mathbb{E}\left[\operatorname{sign}(u^\mathsf{T} g)\operatorname{sign}(v^\mathsf{T} g)\right] = \frac{2}{\pi}\arcsin(u^\mathsf{T} v). \qquad (7)$$

Let $Y = f(XX^\mathsf{T}) \in \mathbb{R}^{n \times n}$ be the elementwise application of the scalar function

$$f(t) = \tfrac{2}{\pi}\left(\arcsin(t) - \tfrac{t}{\gamma(k)}\right). \qquad (8)$$

Lemma 1 in [13] shows that $f(t)$ is a function of the *positive type* on $\mathcal{S}^k$, which by definition means that $Y \succeq 0$ provided $X_i \in \mathcal{S}^k$ for all $i$. The underlying theory is developed in [14].

For $A, Y \succeq 0$ we have that $\operatorname{tr}(AY) \geq 0$. Rearranging terms and applying Grothendieck's identity,

$$0 \leq \operatorname{tr}(AY) = \operatorname{tr}\left(A\frac{2}{\pi}\left(\arcsin(XX^\mathsf{T}) - \frac{XX^\mathsf{T}}{\gamma(k)}\right)\right) \qquad (9)$$

$$\Longleftrightarrow \operatorname{tr}\left(A\frac{2}{\pi}\arcsin(XX^\mathsf{T})\right) \geq \frac{2}{\pi\gamma(k)}\operatorname{tr}(AXX^\mathsf{T}) \qquad (10)$$

$$\Longleftrightarrow \mathbb{E}[\operatorname{IQP}(\operatorname{rrd}(X))] \geq \frac{2}{\pi\gamma(k)}\operatorname{LRP}_k(X), \qquad (11)$$

as claimed. $\qquad\square$

## A.2 MRF to IQP reduction

Using the shorthand $\psi_{i;u} = \psi_i(u)$ and $\theta_{ij;uv} = \theta_{i,j}(u,v)$, the negative energy can be written as a sum of terms $\psi_{i;1}x_i + \psi_{i;0}(1-x_i)$ and of terms

$$\theta_{ij;11}x_ix_j + \theta_{ij;10}x_i(1-x_j) + \theta_{ij;01}(1-x_i)x_j + \theta_{ij;00}(1-x_i)(1-x_j) \tag{12}$$

for every $i, j$, *i.e.* negative energy is a quadratic form over $\{0,1\}^n$, and finding its maximum is precisely the MAP problem. This quadratic form over can be written as $x^\mathsf{T} M x + \beta^\mathsf{T} x + \beta_0$, where

$$M_{i,j} \overset{\text{def}}{=} \theta_{ij;11} + \theta_{ij;00} - \theta_{ij;10} - \theta_{ij;01} \qquad\qquad \text{for } i < j \tag{13}$$

$$\beta_i \overset{\text{def}}{=} \psi_{i;1} - \psi_{i;0} + \textstyle\sum_{j>i}(\theta_{ij;10} - \theta_{ij;00}) + \sum_{j<i}(\theta_{ji;01} - \theta_{ji;00}) \qquad \text{for every } i \tag{14}$$

$$\beta_0 \overset{\text{def}}{=} \textstyle\sum_i \psi_{i;0} + \sum_{i<j}\theta_{ij;00} \tag{15}$$

This in turn can be written more compactly as $x^\mathsf{T}(M' + \mathrm{diag}(\beta))x + \beta_0$, where $M' = (M + M^\mathsf{T})/2$ is taken for symmetry. In summary, MAP in the MRF reduces to maximizing the term left of $\beta_0$ (that which we can control), which is now in a form that differs from IQP only by the domain of $x$.

One can then reduce the problem from the $x \in \{0,1\}^n$ domain to $x \in \{-1,1\}^n$ by a linear change of variables. Given an IQP as in (1) with objective $x^\mathsf{T} A x$ over $x \in \{0,1\}^n$, we can equivalently optimize $[\frac{1}{2}(\tilde{x}+\mathbf{1})]^\mathsf{T} A[\frac{1}{2}(\tilde{x}+\mathbf{1})]$ over $\tilde{x} \in \{-1,1\}^n$. This reduction introduces cross-terms. Define

$$b \overset{\text{def}}{=} \mathbf{1}^\mathsf{T} A + A\mathbf{1} = 2A\mathbf{1} \in \mathbb{R}^n \qquad\qquad b_0 \overset{\text{def}}{=} \mathbf{1}^\mathsf{T} A\mathbf{1} = \tfrac{1}{2}\mathbf{1}^\mathsf{T} b \in \mathbb{R}^n \tag{16}$$

Now, optimizing over $x \in \{-1,1\}^n$, we can fold $b$ and $b_0$ into $A$ by introducing a single auxiliary variable $x_0$ (so the new domain is $x' = (x_0, x)$) and augmenting $A$ to

$$A' = \frac{1}{4}\begin{bmatrix} b_0 & \frac{1}{2}b^\mathsf{T} \\ \frac{1}{2}b & A \end{bmatrix}. \tag{17}$$

The variable $x_0$ must be constrained to $1$, but in practice such a constraint can be ignored up until we output a final solution, because negating all of $x$ has no effect on the IQP objective.

## A.3 Additional figures

Figure 4 shows empirical histograms of objectives of random roundings from an $\mathrm{LRP}_k$ solution.

Figure 4: Distribution of the value of random roundings across problem instances and ranks. From top to bottom, rows vary across $k = 2, 4, 8$. From left to right, columns show: (1) random $A$; (2) a pairwise distance matrix formed by MNIST digits 4 and 9; (3) an instance from **seg**; (4) an instance from **dbn**. The range of the x-axis is identical in each column.