[Reviews · NeurIPS 2014]

Submitted by Assigned_Reviewer_10

This paper studies the technique of "low-rank relaxations" for MAP inference.

The idea is to solve the problem $max_{x\in +-1} x^T A x$ using a relaxed form $max_{X\in \R^{n\times k}} tr(X^T A X)$ where the second maximization is constrained so that rows of $X$ have unit length. The value of $k$ can be varied to get different kinds of approximation. Solutions to the second problem are turned into solutions to the first using "randomized rounding" in which the rows of $X$ are projected onto a random vector and rounded to plus or minus 1.

This idea is not new but the present paper contributes theoretical bounds which guide the choice of $k$. The mathematics was interesting to me and I did not find any mistakes, although I did not check the proofs of any of the theorems. The paper's conclusion is that $k$ does not have to be very large to get good results.

The relaxation-rounding ("R^3") technique was compared to Mean Field and Gibbs on synthetic data with good results. I especially liked the argument in Figure 2 to show that R^3 can avoid some local optima which trap Gibbs and MF.

I also liked the emphasis on the flexible interface provided by the algorithm, allowing different treatment of sparse and dense weight matrices for example (Introduction, penultimate paragraph).

I was not clear on how novel the R^3 algorithm is, apparently from section 6 paragraph 3, similar algorithms have been presented before, but with different emphasis and no theory. It would be good to be able to convey this earlier in the paper, and tell the reader explicitly if "R^3" is a new term that is being coined here...

End of 4.3: "every variables"

There is an argument in the appendix that we need only consider MRFs with pairwise interactions since an auxiliary variable can be introduced to simulate singleton factors. If the auxiliary variable has the wrong value in a solution, simply flip the sign of all variables, and the objective will be preserved due to sign-symmetry of the pairwise-only MRF. It would be nice to see this argument appear early in the paper since otherwise some readers won't realize that the algorithm is fully general.

Quality: I think the paper is of high quality.

Clarity: It is clear and I enjoyed reading it and learning some new mathematics.

Originality: The paper is not terribly original, but seems to provide some important results on an algorithm of interest.

Significance: I think it is reasonably significant. MAP is not my favorite problem, but it's useful in some areas, and studying it gives rise to interesting mathematics which may have wider applicability. I think the paper will be interesting to Bayesians as well as MAP people.
Summary: Interesting paper with important synthesis and analysis of contemporary optimization techniques.

Submitted by Assigned_Reviewer_20

The authors have studied the effect of the rank on low-rank semi-definite program based relaxations of an integer quadratic program problem through a theoretical exposition. The authors also analyse the use of R^3 in practice. The paper is well written and the authors demonstrate a clear understanding of their contributions in the context of the surrounding literature. Indeed, they acknowledge the use of IQP relaxation solved by low-rank relaxation by other authors, but point out that other papers have not studied the basis on which the approximations hold. I also thought that the connections to Gibbs sampling and mean field variational approximations were informative and useful. The paper is clearly written and would be of interest to the NIPS community in my opinion.
Summary: Unfortunately I do not feel qualified to offer a strong opinion on the significance of this paper as it falls outside of may area of expertise. The presentation was clear and precise, and I did not find any mistakes in the paper.

Submitted by Assigned_Reviewer_42

This paper focuses on MAP inference on binary MRFs with pairwise potential functions. This problem is casted as integer quadratic programming (IQP) in this paper.

This is a solid paper. It analyzes the low-rank relaxations that interpolate between the discrete MAP problem and its full-rank semidefinite relaxation, followed by randomized rounding. I found the bounds given in this paper not only have technical depth but also give interesting insight to low-rank relaxation and rounding effect.

The authors also gave simple algorithms to optimize the low-rank objective which bears similarity to Gibbs and mean field approximations.

My only complain about this paper is that in their experiments why not comparing their methods with BP which is also easy for parallelization (using a parallel update schedule) and has shown to work better than mean field approximation.
Summary: This is a solid paper that provides new theoretical analysis of the effect of low rank approximation and rounding and inference algorithms that easy to implement. The experiments can be improved by including comparison with BP results but this is not fatal.
Author Feedback
Author rebuttal: Our thanks to the reviewers for their thoughtful and constructive feedback.

Reviewer #1 suggests early in the paper clarifying the new algorithm name R^3 versus similar algorithms in previous work and mentioning universality due to the sign-flipping argument. We are happy to adopt these helpful comments to improve clarity.

Reviewer #2: thank you for commenting on matters of clarity and thoroughness. We are glad to hear that you liked the paper.

Reviewer #3 suggests an additional comparison to BP, and we agree this would make the experiment suite more complete. In preliminary experiments, we found BP to under-perform when given the same computation budget, since BP messages are more expensive to compute (especially in edge-dense models). Still, a more thorough investigation is needed to reach a definitive conclusion.